# In-Situ Observation of Lüders Band Formation in Hot-Rolled Steel via Digital Image Correlation

**Hai Qiu \*, Tadanobu Inoue** and **Rintaro Ueji**

Research Center for Structural Materials, National Institute for Materials Science, 1-2-1 Sengen, Tsukuba, Ibaraki 305-0047, Japan; Inoue.Tadanobu@nims.go.jp (T.I.); Ueji.Rintaro@nims.go.jp (R.U.)
**\*** Correspondence: QIU.Hai@nims.go.jp

**Abstract:** Although the Lüders yield phenomenon has been investigated for more than 150 years, some understanding of Lüders band formation lack substantial support from experimental evidence. In-situ observation of Lüders band formation in hot-rolled steel experimentally clarified the following facts: (i) When stress reaches the true upper yield stress, the Lüders band begins to nucleate. True upper yield stress is greater than nominal upper yield stress. (ii) Gross stress concentration promotes the Lüders band formation, and the size of the gross stress concentration region determines the initial width of the Lüders band. (iii) The Lüders band nucleates far ahead of the gross yield point.

**Keywords:** Lüders band; Lüders strain; polycrystalline steel; tensile test; digital image correlation

---

## 1. Introduction

The transition from elastic to plastic deformation in low carbon steels and mild steel is characterized by a material instability known as the Lüders deformation phenomenon whose macroscopic deformation is inhomogeneous. In a uniaxial tension test, this instability shows a typical stress-strain curve as illustrated in Figure 1 (black line). It is usually recognized that localized plastic deformation in the form of a band, denoted as a Lüders band, starts with a sudden stress drop. Subsequently, the Lüders band propagates through the whole gauge length of the specimen, while the global stress remains essentially constant. The stress plateau in the stress-strain curve reflects the propagation process. The initial stress peak and the level of the stress plateau are, respectively, the upper yield stress ($\sigma_{up.ys.ob}$) and lower yield stress ($\sigma_{L.ys.ob}$).

Lüders band formation is the beginning of plastic instability. Two questions remain: under what conditions and when does the Lüders band nucleate? Previous studies have given physical interpretations of microscopic and macroscopic views. Cottrell and Bilby [1] reported that the Lüders band formation in iron was accounted for by the pinning of dislocations by the solute interstitials such as C and N atoms, which tend to form atmospheres around them. Onodera et al. found that the Cottrel atmosphere did not agree with an alloy Al-4Cu-0.5Mg-0.5Mn [2]. Hahn [3] proposed another model in which the dominant mechanism of Lüders band formation is attributed to rapid dislocation multiplication.

On a macroscopic scale, the upper yield stress represents the stress for the unlocking, creation, or rapid multiplication of mobile dislocations [4]. Van Rooyen recognized the upper yield stress as a critical macroscopic parameter and suggested that, when macroscopic applied stress reaches the upper yield stress, a Lüders band begins to form [4,5]. The upper yield stress is greater than the experimental upper yield stress [4,5]. The Van Rooyen model well depicts the effects of grain size and strain rate on the Lüders strain. However, the predicted strain distribution across a Lüders front is inconsistent with the experimental result [6]. Schwab and Ruff [7] believed that the stress plateau in the nominal stress-strain curve does not truly reflect the material behavior. The experimental upper and lower yield

stresses ($\sigma_{up.ys.ob}$ and $\sigma_{L.ys.ob}$, respectively) in Figure 1 are only observed variables, and the true upper and lower yield stresses ($\sigma_{up.ys.tr}$ and $\sigma_{L.ys.tr}$, respectively) are different from them. The stress plateau should be replaced by the red curve, i.e., the red curve reflects the true material behavior. This model is based on three assumptions, and one of them is that $\sigma_{up.ys.tr}$ is greater than $\sigma_{up.ys.ob}$.

The Lüders band is believed to nucleate at the moment of the stress drop, i.e., the macroscopic yield point [8,9]. Nagarajan's work [6] indicated that Lüders band formation takes place close to the macroscopic yield point. All existing models of Lüders band formation are built on assumptions of a lack of experimental evidence and support. The formation time of the Lüders band needs to be further investigated. Several experimental studies have been performed on the Lüders behavior using digital image correlation (DIC) [10–12]. In this study, our aim is to try to answer the two questions through in-situ observation using a digital image correlation (DIC) technique that has been proven useful [9–13].

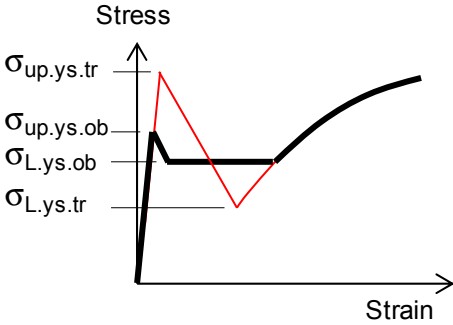

**Figure 1.** Black lines represent a schematic diagram of the stress-strain curve with Lüders deformation. Red lines represent the Schwab-Ruff model [7].

## 2. Materials and Methods

A commercial hot-rolled steel SM490 was studied. Its chemical composition is 0.16C/0.44Si/1.46Mn (in wt%), and it is composed of ferrite (average grain size 11 μm) and pearlite. Two types of dog-bone-type specimen (Type I and Type II) were used whose specimen sizes are shown in Figure 2. The front surface was sprayed with white and black paint to make speckles for DIC analysis. The x axis is the tension direction. Three pieces of Type I specimen and two pieces of Type II specimen were machined. All the tension tests were performed at room temperature and at a crosshead speed of 0.01 mm/s. The global strain ($\varepsilon_{x.g}$) was obtained using an extensometer attached to the back surface whose gauge length (GL) is 30 mm. The deformation process on the front surface was recorded with a CCD camera (maximum frame rate, 15 fps) at a constant time interval of 500 ms. The area observed by 2D-DIC for Type I specimen is 20 mm × 9 mm, and the area for Type II specimen is 30 mm × 9 mm. Their positions are shown in Figure 2. The local strain and strain rate along the x-axis ($\varepsilon_{x.local}$ and $\dot{\varepsilon}_{x.local}$, respectively) in these areas were determined using 2D-DIC (a software VIC-2D, produced by the Correlated Solution, Inc.). The 2D-DIC measurement was within the global elastic region in which applied stress ranged from zero to $\sigma_{up.ys.ob}$. The result of DIC analysis is sensitive to the DIC parameter [14], and, thus, three conditions of DIC analysis were used in the correlation operation: ① subset size, 9 pixels × 9 pixels (248 μm × 248 μm), step, 5 pixels (138 μm), ② subset size, 21 pixels × 21 pixels (579 μm × 579 μm), step, 5 pixels (138 μm), and ③ subset size, 41 pixels × 41 pixels (1.13 mm × 1.13 mm), step, 5 pixels (138 μm).

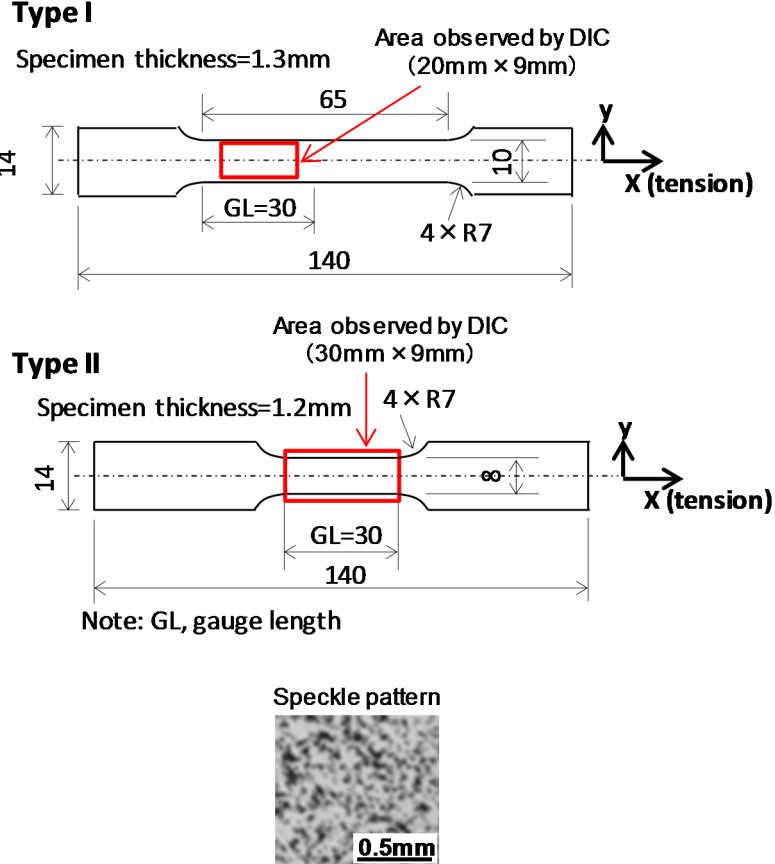

**Figure 2.** Two types of tension specimen (Type I and Type II) and speckle patterns on the front surface of the specimen.

Van Rooyen [5] believed that the local stress can be related to the local strain in terms of a basic stress-strain curve. Van Rooyen [4] suggested that the basic stress-strain curve can be obtained from the experimental stress-strain curve with Lüders deformation, as shown in Figure 3. The elastic region of the basic stress-strain curve is directly from the experimental data, and the plastic region (strain hardening region) is given by the equation $\sigma = k\varepsilon^n$ ($\sigma$, true stress; $\varepsilon$, true strain), where constants $k$ and $n$ are obtained from plotting the strain hardening region in the experimental stress-strain curve. The values of $k$ and $n$ are, respectively, 566 MPa and 0.060 for specimen-1 shown in Figure 4. Using Van Rooyen's approach, we obtained the basic stress-strain curve of the steel SM490 from the experimental stress-strain curve. The local strain was directly obtained from the DIC analysis, and its corresponding local stress was determined according to the basic stress-strain curve.

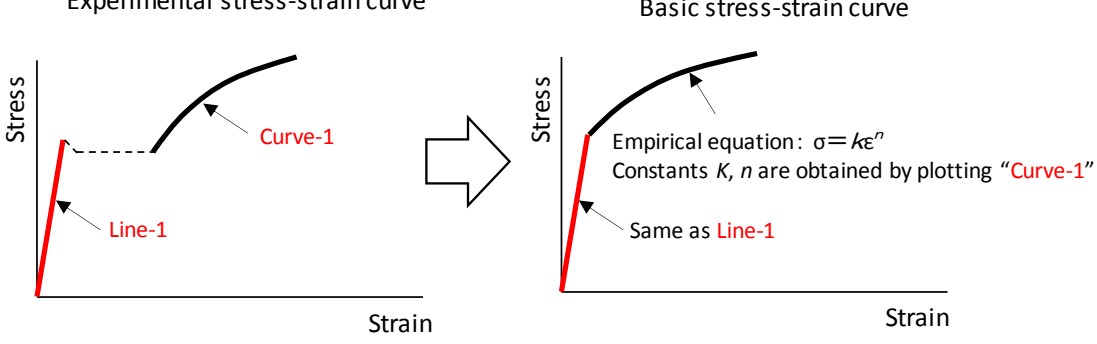

**Figure 3.** Schematic representation of a basic stress-strain curve [4].

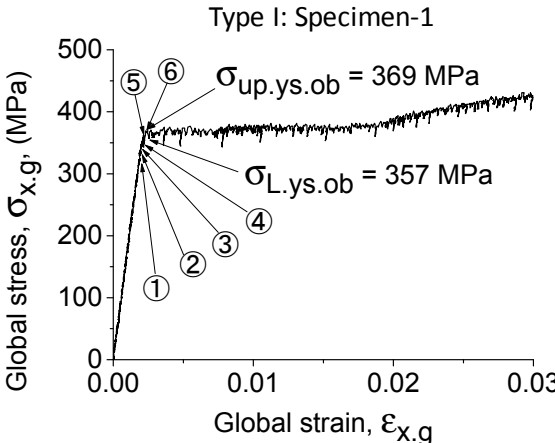

**Figure 4.** Experimental stress-strain curve. The global nominal strain was measured using an extensometer with a gauge length (GL) of 30 mm.

## 3. Results and Discussion

Three tension tests on Type I specimen were performed. The observed area partially covered the gauge length (see Figure 2). Because the initiation site of Lüders band was beyond the observed area, two tests failed to capture the initiation of Lüders band. Only one test succeeded in capturing the band initiation. Its experimental global stress-strain curve is given in Figure 4. We defined the first peak point as the macroscopic upper yield point in this study.

Images ① to ⑥ shown in Figure 4 were selected. Figure 5a shows the $\varepsilon_{x.local}$ evolution from image ① to image ⑥, corresponding to the applied stress levels of 318 MPa ($0.86\sigma_{up.ys.ob}$), 332 MPa ($0.90\sigma_{up.ys.ob}$), 345 MPa ($0.93\sigma_{up.ys.ob}$), 351 MPa ($0.95\sigma_{up.ys.ob}$), 364 MPa ($0.99\sigma_{up.ys.ob}$), and 369 MPa ($\sigma_{up.ys.ob}$), respectively (see Figure 4). The DIC parameter used is a subset size of 9 pixels × 9 pixels (248 μm × 248 μm) and step 5 pixels (138 μm). A region is enclosed by a red circle in Figure 5a. Strain concentration has occurred at a stress level of $0.86\sigma_{up.ys.ob}$ (image ①), and a small core (shown by an arrow) is visible. At $0.90\sigma_{up.ys.ob}$ (image ②), the core inclines and changes to an elliptical shape. This shows a trace of plastic propagation. In image ③, two cores (marked A and B) are visible. Core A propagates along the arrow toward the lower right side, forming Band 1. Core B goes in the opposite direction to form Band 2. At $0.95\sigma_{up.ys.ob}$ (image ④), Band 1 was completely formed. The band has an angle of 55° with respect to the tension direction. Core B continues to propagate and completely crosses the specimen width at 0.99 $\sigma_{up.ys.ob}$ (image ⑤). When the applied stress reaches the $\sigma_{up.ys.ob}$, the strain within the Lüders band, i.e., the Lüders strain, reaches its maximum (image ⑥). The Lüders band forms in a very short time. For example, the formation process of Band 1 from image ② to image ④ took only 8 s.

The strain rate fields corresponding to each image in Figure 5a are given in Figure 5b. Although strain concentration has occurred in Figure 5a images ①–④, and a Lüders band has even formed in image ④, serious concentration does not appear in the strain rate field (see Figure 5b, images ①–④). Subsequently, in images ⑤ and ⑥, a strain rate band appears. The strain rate within the band along the specimen width is inhomogeneous. The center has a lower strain rate, and both sides near the end of the specimen have a higher strain rate. As we know, plastic strain concentration first occurs in the center (cores A and B), and then the plastic flow crosses from there to both ends of the specimen. The increasing strain rate distribution within the band shows the propagation process, like a river flowing from a higher place to a lower place. The flow speed increases gradually. The strain rate of bulk material was directly derived from the evolution of global strain with time. Its value within the elastic region is of the order of $10^{-5}$ s$^{-1}$, and the maximum strain rate within the Lüders band is of the order of $10^{-3}$ s$^{-1}$. This shows that the Lüders band deforms at a very fast speed (100 times the speed in bulk material).

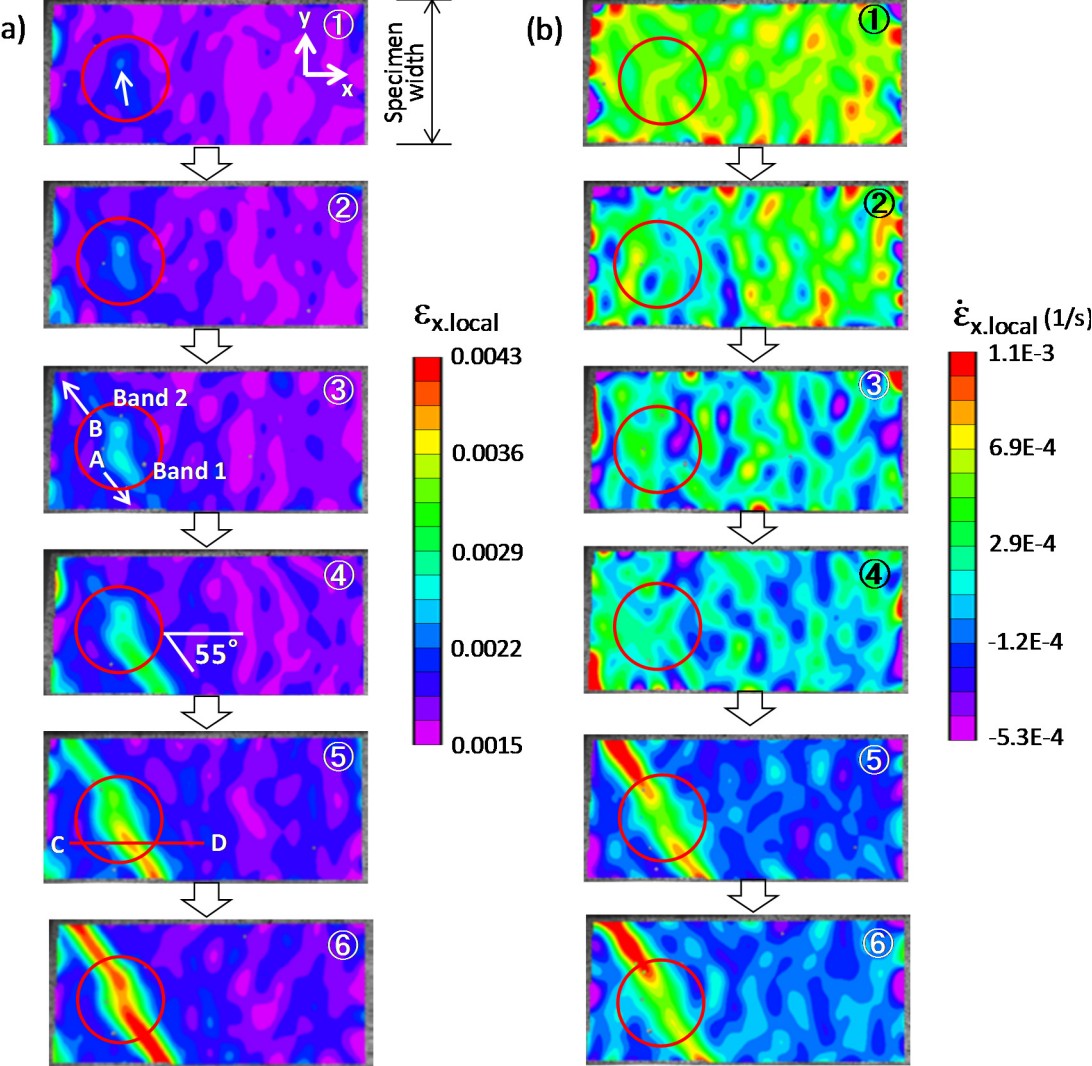

**Figure 5.** (**a**) Evolution of the strain field. (**b**) Evolution of the strain rate field. Global stress: ① $0.86\sigma_{up.ys.ob}$ (318 MPa), ② $0.90\sigma_{up.ys.ob}$ (332 MPa), ③ $0.93\sigma_{up.ys.ob}$ (345 MPa), ④ $0.95\sigma_{up.ys.ob}$ (351 MPa), ⑤ $0.99\sigma_{up.ys.ob}$ (364 MPa), ⑥ $\sigma_{up.ys.ob}$ (369 MPa). Time interval: ① to ②, 4 s, ② to ③, 5.5 s, ③ to ④, 2.5 s, ④ to ⑤, 1 s, ⑤ to ⑥, 0.5s. DIC parameters: subset size, 9 pixels × 9 pixels (248 μm × 248 μm), step, 5 pixels (138 μm).

The images ①–⑥ in Figure 4 were also dealt with the following DIC parameters: (1) subset size 21 pixels × 21 pixels (579 μm × 579 μm) and step 5 pixels (138 μm), and (2) subset size 41 pixels × 41 pixels (1.13 mm × 1.13 mm) and step 5 pixels (138 μm). The corresponding evolution of local strain field is shown in Figure 6. Compared with Figure 5, it can be seen that the sensitivity of a large subset size is lower than that of a small subset size (see the region in the red circle). The DIC parameter of subset size 9 pixels × 9 pixels and step 5 is rational.

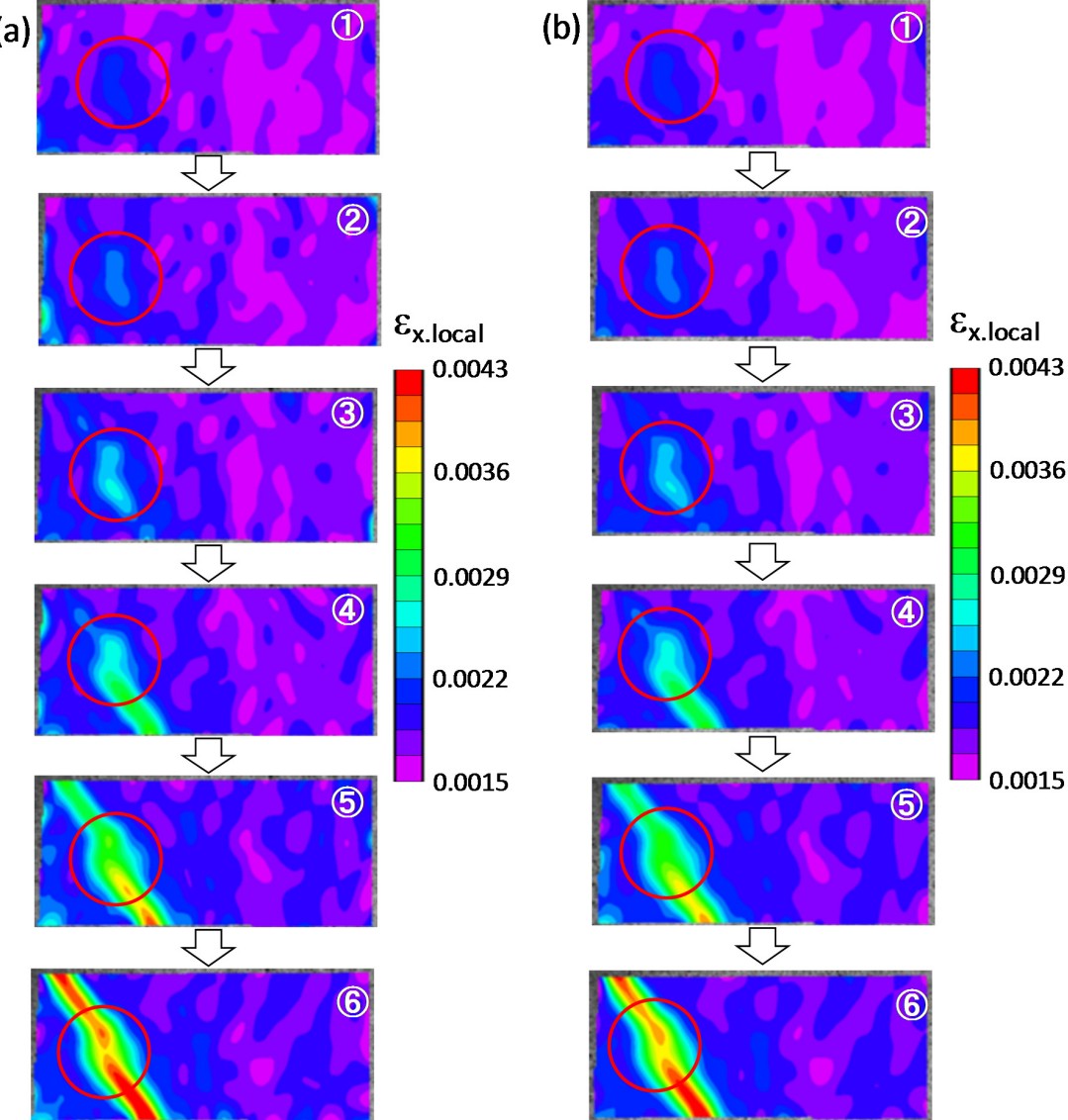

**Figure 6.** Evolution of the strain field. DIC parameters: (**a**) subset size, 21 pixels × 21 pixels (579 μm × 579 μm), step, 5 pixels (138 μm), and (**b**) subset size, 41 pixels × 41 pixels (1.13 mm × 1.13 mm), step 5 pixels (138 μm). Each image corresponds to the image in Figure 5.

Two tension tests on Type II specimen were carried out. Their experimental global stress-strain curves are given in Figures 7a and 8a, respectively. In Figure 7a, stress fluctuation on the yield plateau is great. The nominal upper yield stress (point ④) is not the largest on the yield plateau. The $\varepsilon_{x.local}$ field and $\dot{\varepsilon}_{x.local}$ field evolution from image ① to image ④ is shown in Figure 7b. A Lüders band begins to form at the shoulder of the specimen at the stress level of $0.927\sigma_{up.ys.ob}$ (Figure 7b—image ②).

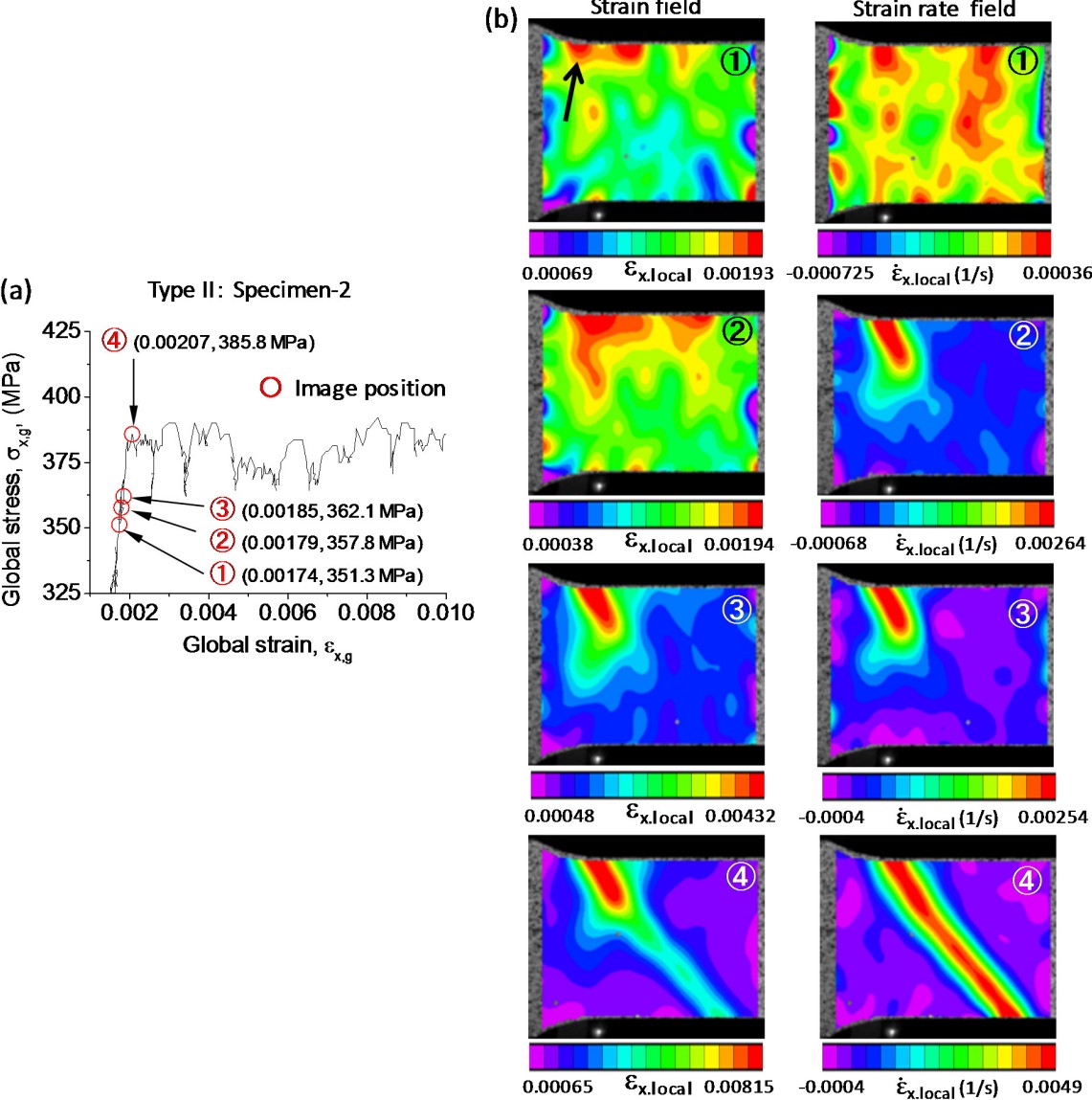

**Figure 7.** (**a**) Image position in the stress-strain curve. (**b**) Strain field and strain rate field of image ① to image ④. ① Global stress level $0.911\sigma_{up.ys.ob}$, ② global stress level $0.927\sigma_{up.ys.ob}$, ③ global stress level $0.939\sigma_{up.ys.ob}$, and ④ global stress level $\sigma_{up.ys.ob}$. Time interval: ① to ②, 0.5 s, ② to ③, 0.5 s, ③ to ④, 2.5 s. DIC parameters: subset size, 9 pixels × 9 pixels (246 μm × 246 μm), step, 5 pixels (137 μm).

In Figure 8a, the stress-strain curve is similar to that shown in Figure 1. The $\varepsilon_{x.local}$ field and $\dot{\varepsilon}_{x.local}$ field evolution from image ① to image ④ is shown in Figure 8b. It can be seen that the formation of a Lüders band takes place at the stress level of $0.849\sigma_{up.ys.ob}$ (Figure 8b—image ②). The initiation site is indicated by the arrow at image ②. The observation (Figures 5, 7 and 8) shows that, at the upper yield point, the Lüders band has grown large enough, passing through the specimen width. In Figure 5b, the strain rate field is not sensitive to identify the band. However, in Figures 7b and 8b, the strain rate field is more sensitive than the strain field in identifying a band. The reason for it is unclear. It is better to identify a band using both strain and strain rate fields.

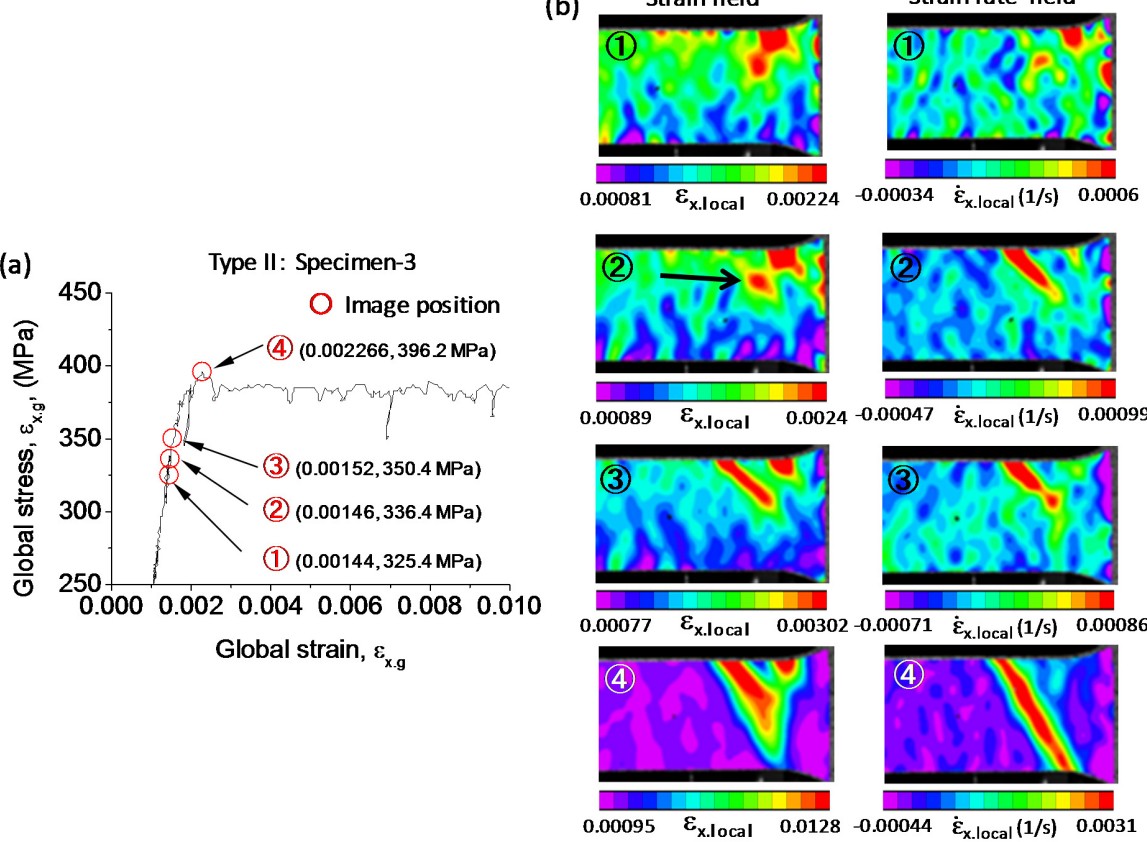

**Figure 8.** (**a**) Image position in the stress-strain curve. (**b**) Strain field and strain rate field of image ① to image ④. ① Global stress level $0.821\sigma_{up.ys.ob}$, ② global stress level $0.849\sigma_{up.ys.ob}$, ③ global stress level $0.884\sigma_{up.ys.ob}$, and ④ global stress level $\sigma_{up.ys.ob}$. Time interval: ① to ②, 0.5 s, ② to ③, 0.5 s, ③ to ④, 12.5 s. DIC parameters: subset size, 9 pixels × 9 pixels (246 μm × 246 μm), step, 5 pixels (137 μm).

The scatter among the three experimental stress-strain curves, especially in the region of Lüders yield plateau, is great (see Figures 4, 7a and 8a). The yield tooth is visible only in Figure 8a. Yuzbekova et al. [15] suggested that the stress serration on the yield plateau is related to the nucleation and propagation of a deformation band. If the Lüders band is developed progressively instead of being formed abruptly, the yield tooth will not show up. The steel SM490 is a commercial steel produced by hot-rolling. It is composed of multiple phases (ferrite and pearlite). The ferrite grain size is not uniform, and the pearlite prefers to distribute along the rolling direction. Moreover, precipitates and inclusions are present. The role of the strong heterogeneity in the microstructure on the stress serration should be investigated in the future.

Strain concentration is related to the formation of a Lüders band. Since the initiation of a Lüders band is similar in Figures 5, 7 and 8, we only focus on Figure 5 in the following discussion. To quantitatively analyze the strain concentration, we plotted a line parallel to the x-axis (CD line) passing through a strain concentration core in the strain field of Figure 5a—⑤ and extracted the $\varepsilon_{x.local}$ along this line. The solid line in Figure 9 represents its $\varepsilon_{x.local}$ distribution. Numbers ①–⑥ in Figure 9 correspond, respectively, to the images ①–⑥ in Figure 5. A peak appears around position x = −14.6 mm, and the peak value increases, as applied stress increases. The dashed horizontal line shows a strain level of 50% of the peak value.

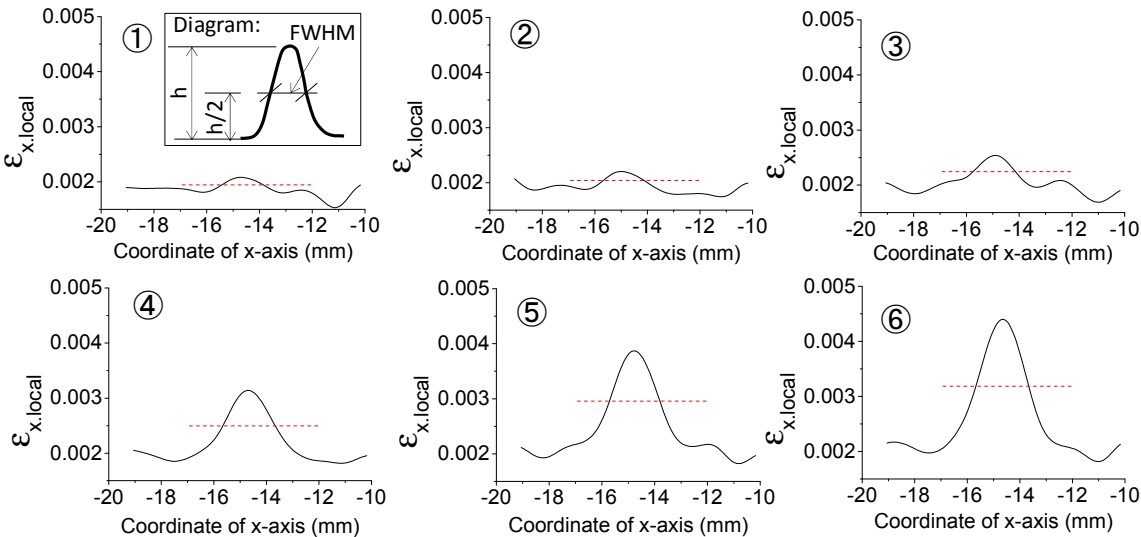

**Figure 9.** Local strain distribution along the CD line in Figure 5. The dashed line represents half of the peak value. ① to ⑥ correspond, respectively, to each image in Figure 5.

Applied force simultaneously makes the bulk material and local regions produce global strain ($\varepsilon_{x.g}$) and local strain ($\varepsilon_{x.local}$), respectively. We define a factor, the ratio of the maximum local strain to global strain, $f_\varepsilon = \varepsilon_{x.local.max} / \varepsilon_{x.g}$, to evaluate the extent of the strain concentration, where $\varepsilon_{x.local.max}$ is the maximum local strain (peak value in Figure 9). The evolution of $f_\varepsilon$ as a function of normalized applied stress ($\sigma_{x.g} / \sigma_{up.ys.ob}$) is shown in Figure 10a.

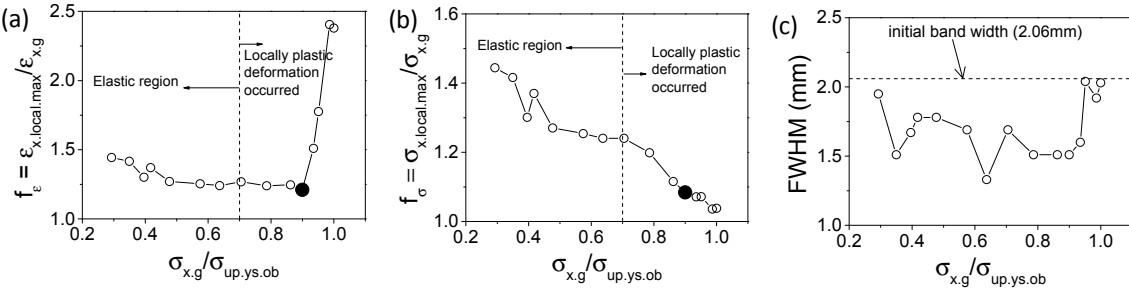

**Figure 10.** The dependence of (**a**) the strain concentration factor, (**b**) the stress concentration factor, and (**c**) the size of the stress concentration region on the normalized applied stress.

The critical strain ($\varepsilon_{cri}$) is calculated using $\sigma_{up.ys.ob} / E$ (Young's modulus E = 210 GPa), beyond which the bulk material enters the plastic region. For the steel used, the $\varepsilon_{cri}$ is equal to 0.00176, which is used to determine the boundary between elastic and plastic regions. Figure 10a shows that the global applied stress level of $0.70\sigma_{up.ys.ob}$ is an elastic-to-plastic threshold value, below which both the bulk material and the local region completely and elastically deform and above which some plastic local regions gradually appear. Within the applied stress range of $0.3$–$0.9\sigma_{up.ys.ob}$, $f_\varepsilon$ slightly decreases with applied stress, i.e., the extent of the strain concentration slightly decreases. The solid black circle in Figure 10a is a turning point, and when applied stress exceeds that point, $f_\varepsilon$ significantly increases in a short period of time. In-situ observation in Figure 5 shows that only strain concentration occurs before the stress level of $0.90\sigma_{up.ys.ob}$. At $0.90\sigma_{up.ys.ob}$, a Lüders band begins to form, which is followed by rapid propagation. The suddenly increased local strain in Figure 10a is attributed to the formation and propagation of the Lüders band.

Strain concentration is directly related to stress concentration. We define the stress concentration factor, $f_\sigma$, by $\sigma_{x.local.max} / \sigma_{x.g}$ (where $\sigma_{x.local.max}$ is the maximum local stress and $\sigma_{x.g}$ the global stress) and convert $f_\varepsilon$ into $f_\sigma$. Figure 10b shows the evolution of $f_\sigma$ against normalized applied stress. The bulk

material is in an elastic state below $\sigma_{up.ys.ob}$, while the stress concentration region is in an elastic state only below $0.70\sigma_{up.ys.ob}$. In the elastic region, stress ($\sigma$) and strain ($\varepsilon$) have a linear relation, as $\sigma = E\varepsilon$. We assume that the stress concentration region has the same Young's modulus as the bulk material. Therefore, $f_\sigma$ is equal to $f_\varepsilon$ in the range of $0$–$0.70\sigma_{up.ys.ob}$. In the plastic region, the relationship between $\sigma$ and $\varepsilon$ is expressed by $\sigma = k\varepsilon^{\,n}$ ($k$ and $n$ are constants and $n$ is smaller than one) [16], and, thus, $f_\sigma$ is not equal to $f_\varepsilon$. Figure 4 shows that the strain-hardening capacity of the steel used is weak. The local stress in the plastic stress concentration region is obtained by means of the basic stress-strain curve introduced in the Materials and Methods section. Figure 10b indicates that the $f_\sigma$ decreases with the applied stress.

The Lüders band nucleates at the stress concentration site. In the following, we will investigate the correlation of the Lüders band width with the size of the stress concentration region. As shown in Figure 9, there is no distinct boundary between the concentration region and its adjacent region. Therefore, it is difficult to determine the size of the stress concentration region. It is convenient to use full width at half maximum (FWHM) as a measure of size, which is schematically shown in Figure 9—①. The size of the stress concentration region (FWHM) against normalized applied stress is given in Figure 10c. The FWHM varies within 1.33–2.04 mm. Around the $\sigma_{up.ys.ob}$, the FWHM ranges from 1.92 to 2.04 mm. Band 1 formed completely at $0.95\sigma_{up.ys.ob}$, and the band front has a clear boundary with the matrix (see Figure 5a—④). We plot a horizontal line, which intersects with the two band fronts, and measure the distance between the two intersection points to give the initial band width. The measured band width is 2.06 mm, which agrees with the FWHM at $0.95\sigma_{up.ys.ob}$. This indicates that the initial Lüders band width is dependent on the size of the stress concentration region. Previous studies showed that the band width generally ranges from 1 to 15 mm [17].

We briefly summarize previous studies on Lüders band formation as follows.

i     Band nucleation criterion [5,7]. True upper yield stress ($\sigma_{up.ys.tr}$) is a critical value. When stress reaches it, a band starts to nucleate. The $\sigma_{up.ys.tr}$ is not the nominal upper yield stress ($\sigma_{up.ys.ob}$) obtained from the experimental nominal stress-strain curve, and it is difficult to be measured directly by experiment. $\sigma_{up.ys.tr}$ is greater than $\sigma_{up.ys.ob}$.

ii    Band nucleation site [5,8,13]. A band nucleates at the stress concentration site. The initial band is generally at the shoulder of the specimen where great stress concentration occurs.

iii   Time point of band nucleation [6,8,9]. A band nucleates at the nominal upper yield stress point or close to it.

The solid black circle in Figure 10b indicates the starting point of band nucleation, and, thus, the local stress at this point is the $\sigma_{up.ys.tr}$ (394 MPa). Figure 5 shows that macroscopic stress concentration regions provide sites for easy Lüders band formation. The Lüders band nucleates at $f_\sigma = 1.07$ (or $f_\varepsilon = 1.21$). The importance of plastic strain to the nucleation of the Lüders band was recognized in the 1950s [18,19]. Vreeland et al. [18] observed a pre-yield phenomenon prior to the onset of yielding and found the plastic micro-strain to be of the order of 30 $\mu\varepsilon$. The plastic strain on a macroscale is regarded to be a prerequisite of Lüders band generation [19]. The plastic strain at the onset of band nucleation in this study was 365 $\mu\varepsilon$, which is much greater than Vreeland's data. Vreeland likely underestimated the real plastic strain because of the imprecision measurement technique at that time. At the nominal upper yield stress point, the yield band on a well-polished surface is wide enough to be visible even with the naked eye. If the measurement has enough precision, one will find that a yield band was formed prior to the nominal upper yield stress point. Our observation shows that a band nucleates ahead of the nominal upper yield stress point.

## 4. Conclusions

We performed in-situ observation of a Lüders band formation process. Based on these experimental results, we reached the following conclusions.

(1)    The extent of the stress concentration decreases with the increase in applied stress.
(2)    The local stress at which an initial Lüders band begins to form is greater than the nominal upper yield stress.
(3)    Global stress concentration promotes the Lüders band formation.
(4)    The size of the global stress concentration region determines the initial Lüders band width.
(5)    The Lüders band nucleates far ahead of the global yield point.

**Author Contributions:** H.Q., T.I. and R.U. contributed equally to this work. All authors have read and agreed to the manuscript.

**Funding:** This research received no external funding.

**Conflicts of Interest:** The authors declare no conflict of interest.

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
