# Peer review of "In-Situ Observation of Lüders Band Formation in Hot-Rolled Steel via Digital Image Correlation"

_metals, doi:10.3390/met10040530_

Round 1

Reviewer 1 Report

The paper deals with a challenging question of the Lueders band formation with regard to the mechanical response of the machine-specimen. New data have been obtained. The manuscript thus has a good potential, but some questions should be addressed before accepting it for publication.

Materials and methods.

- Indicate the camera acquisition rate (frames per second). As the strain rate is further calculated, the acquisition rate is of great importance.

- line 65. Besides the crosshead speed, indicate the nominal applied strain rate. Moreover, the gauge length of the sample should also be given explicitly.

- Explain how the local stress was calculated.

Results and discussion

- Figure 3. The Lueders plateau is accompanied with stress serrations with an amplitude comparable with the difference between the upper yield stress and the estimate suggested by the authors for the lower yield stress. It cannot be said if the latter does exist. Very likely, it does not correspond to the lower yield stress in terms of the scheme of Figure 1 but to the unloading produced by the first stress drop. In other words, the upper and lower yield stress values may be equal for this material. This ambiguity in the possible interpretation of the data must be mentioned. Accordingly, it’d be correct to omit the tentative comparison in the paragraph starting from Line 171. This paragraph is even more so unclear because of the missing explanation of the method used to assess the local strain rate (see the above comment).

In this context, discuss the suggestion made in D. Yuzbekova et al, Intern. J. Plasticity 96 (2017) 210: ““The yield plateau is related to nucleation and fast propagation of a deformation band from one specimen edge to the other (Fig. 4), in agreement with the generally accepted Lueders mechanism (Schwab and Ruff, 2013; Antolovich and Armstrong, 2014). The first two frames seize the process of progressive nucleation of the band. This observation might explain the absence of a yield tooth that is known to often precede the Lueders plateau.” It is probable that the equality between the upper and lower yield stress for the studied material is related to such a progressive development of the Lueders band. Note that such a suggestion has two consequences:

- the fact of the early occurrence of the Lueders bands regarding the upper yield stress shouldn’t be considered as a rule but rather as one of the possible scenarios. The discussion and conclusions have to be rewritten in this sense.

- It should be underlined that future investigations are imposed using a material demonstrating a yield tooth.    

- Figure 4 and its description in the text.

The derivative of a variable (here the strain rate) usually reveals heterogeneities clearer than the variable itself. Isn’t the worse visibility of the band in the local strain rate field due to a relatively low acquisition rate?   

Line 105. The value of 10-5 s-1 is given for the strain rate in the bulk of the sample. To which color does it correspond in the snapshots of Figure 4? The corresponding color could be indicated for each frame by an arrow or by an additional circle. Note that this question is related to the second question above (the nominal applied strain rate). If I understand well, it is something like 0.01 mm/s \div 60 mm (I estimated the length in Figure 2) = 1.6 10-4.

Starting from Line 137, it is difficult to comment before the authors explain how the local stress was assessed.

Reference [7] should appear after the first occurrence of this citation in the legend to Fig. 1.

Author Response

I greatly appreciate your comments. The answers to your comments are summarized in the attached word file.

Reviewer 2 Report

This article deals with in-situ observation of Lüders band formation in hot-rolled steel via digital image correlation. The analysis is carried out from surface measurements in two-dimensions at room temperature on one tensile test performed one a sample of dog-bone-type.

I found the article not well written and not very clear. Among my (many) concerns I did not find that the topic of the article is properly introduced. The scientific questions that remained open are passed over and not discussed enough to put this work in an adequate context. In addition, several methodological problems can be raised such as, for example, the total absence of any explanation of the DIC method implemented, the fact that the analysis relates only to a single sample (what about it of reproducibility?) or the fact that the theoretical models mentioned are not at all introduced and explained (Schwab-Ruff model). Finally, we could observe that Figure 3 really does not look like Figure 1 at all (e.g. it has almost no strain-softening), which poses many problems for the measurement of parameters and the interpretive framework, from which I fin the conclusion weak.

For all these reasons, I think this article does not deserve to be published in the Metals journal. In addition to my main concerns, I also listed the following issues:

1) The authors write :

"For metals with a body-centered cubic (bcc) structure, the transition from elastic to plastic deformation is characterized by a material instability known as the Lüders deformation phenomenon whose macroscopic deformation is inhomogeneous"

I think it’s a statement that’s just too general. One should at least have discussed about the phenomenon of aging.

2) "Figure 1. Black lines, schematic diagram of the stress-strain curve with Lüders deformation; red lines, Schwab-Ruff model."

Here, I think that a citation must be added.

3) "Van Rooyen recognized the upper yield stress as a critical macroscopic parameter and suggested that when macroscopic applied stress reaches the upper yield stress, a Lüders band begins to form [4,5]. The upper yield stress is greater than the experimental upper yield stress [4,5]."

Why? And what the difference between these two stresses ? Is it just a matter of stress concentration and inhomogeneities ?

4) It is written that :

"Schwab and Ruff [7] believed that the stress plateau in the nominal stress-strain curve does not truly reflect the material behavior."

Why it is the case? Once again the discussion lacks of depth.

5) The authors write:

"All existing models of Lüders band formation are built on assumptions of the lack of powerful support of experimental evidence, and the formation time of the Lüders band needs to be further investigated."

Here I think there is problem in the sentence. Do the authors mean that the model are built despite the lack of experimental evidence?

6) "In this study, our aim is to try to answer the two questions through in-situ observation using a digital image correlation (DIC) technique which has been proven usefuln [9,10]."

I am sorry but I do not see the two questions at stake. The text should be clearer.

7) It is written that:

"The 2D-DIC measurement was within the global elastic region in which applied stress ranged from zero to σ up.ys.ob ."

Here again, I think there is problem in the sentence. Do the authors mean that the 2D-DIC measurement was calibrated, or tested, in the elastic regime?

8) One of my main concern is that figure 3 is very different from figure 1 from which parameters can be defined. How, from figure 3, the authors choose the parameters corresponding to figure 1? On what criteria?

9) Following my main concerns, there are absolutely no detail regarding th DIC method.

10) It is written in the caption of figure 6c:

"Figure 6. The dependence of (a) strain concentration factor, (b) stress concentration factor, and (c) the size of the stress concentration region on the normalized applied stress."

Wouldn't that be the size of the strain concentration region rather than of the stress? If not, how it is measured?

11) "Figure 6(a) shows that the global applied stress level of 0.70σ up.ys.ob is an elastic-to-plastic threshold value, below which both the bulk material and the local region completely elastically deform and above which some plastic local regions gradually appear.”

On the basis of the figure, it is far to be obvious to me. One would have say 0.9 rather the 0.7. Can the authors justify their choices?

12) It is explained that:

"Assuming that the stress concentration region has the same Young’s modulus as the bulk material. Therefore, f σ is equal to f ε in the range of 0-0.70σ up.ys.ob ."

I think that this is what should have been compared in Figure 6.

13) "The front of the Lüders band inclined 55°in this study (see Fig. 4(a)- 4 ). For this angle, the Schwab-Ruff model gives σ up.ys.tr /σ L.ys.tr =1.26 and σ L.ys.ob /σ L.ys.tr =1.16, deducing that σ up.ys.tr =1.09σ L.ys.ob . According to the Schwab-Ruff model, σ up.ys.tr /σ up.ys.ob = 1.09σ L.ys.ob /σ up.ys.ob . Substituting the experimental values, giving σ up.ys.tr /σ up.ys.ob =1.05. The model-estimated σ up.ys.tr /σ up.ys.ob (=1.05) agrees with the experimental one (=1.07), verifying the proposed band nucleation criterion."

The model is absolutely not introduced or discussed.

14) The authors write:

"The plastic strain at the onset of band nucleation in this study was 365 , much greater than

Vreeland’s data. Vreeland probably underestimated the real plastic strain because of the imprecision measurement technique at that time."

In the absence of further analysis, this proposition is truly speculative.

Author Response

I greatly appreciate your comments. Please see the attachment

Reviewer 3 Report

The very first line of the introduction is inaccurate. Not all bcc metals show the yield point and Lüders band phenomena.

“…the lack of powerful support of experimental evidence, and the formation time of the Lüders band needs to be further investigated”: This statement is correct, however, the present work is not the first of its kind to study the nucleation and propagation of Lüders bands using experimental full-field measurements (e.g. DIC). In fact, there have been several experimental studies that are based on this exact idea, for instance, see [1-3].

The speckle pattern inset in Figure 2 needs a scale bar. Is the DIC area of interest 248x248 (um) or 9x20 (mm)?

What is the point of showing the strain rate fields? How does a strain rate field help with the understanding of the mechanisms associated with Lüders band formation?

On what basis have the authors selected the area shown in Figure 2 as their DIC area of interest? Have the authors used any kind of stress concentrator to limit the location of the plasticity initiation to that specific area? How do you make sure that the Lüders bands do not initiate elsewhere within the gauge area?

It looks like there has only been one experiment carried out in this work. From an experimental perspective, a highly uncertain phenomenon such as inhomogeneous yielding in steels, cannot be fully understood with a single experiment.

There is a serious lack of critical information regarding the DIC analyses in this work. DIC results, especially in cases where strong deformation heterogeneities exist [4], are highly sensitive to the DIC parameters used (i.e. subset, step, strain filter size). The measurement noise floor also highly depends on these parameters. Unless a thorough and systematic identification of optimal DIC parameters is performed, the results cannot be trusted. There is an obvious lack of such systematic analyses in the present study, making the entire conclusions based on questionable assumptions.

[1] Identification of elasto-visco-plastic parameters and characterization of Lüders behavior using digital image correlation and the virtual fields method. Mechanics of Materials 2008; 40: 729.

[2] The Influence of Specimen Thickness on the Lüders Effect of a 5456 Al-Based Alloy: Experimental Observations. Metals 2016; 6: 120.

[3] Studying the Influence of the Loading System on the Spatial-Time Inhomogeneity of Inelastic Deformation in Metals by Analyzing Strain and Temperature Fields. AIP Conference Proceedings 2016; 1785: 030032

[4] Experimental determination of Representative Volume Element (RVE) size in woven composites. Optics and Lasers in Engineering 2017; 90: 59.

Author Response

(The authors gave the same response as above.)

Round 2

Reviewer 1 Report

The authors have answered to formal questions concerning details of experiment and analysis. However, some principal questions have been circumvented. Besides, the answers uncover the misunderstanding by the authors of some basic and methodological issues. However, as the experimental results are interesting, I do not suggest rejecting the paper at this stage, on condition that the following items will be reconsidered.

  1. Mistake in the understanding of the mechanical test.

Old question 2: “- line 65. Besides the crosshead speed, indicate the nominal applied strain rate. Moreover, the gauge length of the sample should also be given explicitly.”

Authors’ answer: In the present study, due to the capacity of the tension equipment, we only controlled

the crosshead speed and cannot controlled the applied strain rate...

And

Old question 6: “Line 105. The value of 10-5 s-1 is given for the strain rate in the bulk of the sample. To which color does it correspond in the snapshots of Figure 4? The corresponding color could be indicated by an arrow or by an additional circle for each frame. Note that this question is related to the second question above (the nominal applied strain rate). If I understand well, it is something like 0.01 mm/s \div 60 mm (I estimated the length in Figure 2) = 1.6 10-4.”

Authors’ answer (see also a scheme in metals-748041-coverletter.pdf): The strain rate (the order of 10-5 s-1) in the bulk sample is an average value of the whole strain rate contour, not a specific region. The schematic diagram of the tension system is shown in the following figure. One side of the specimen was fixed, and the other side was tensioned. The average velocity over the whole specimen is V1/2. The length of the parallel part of specimen is 65mm. The average strain rate is V1/2/65. If we take V1= V0=0.01mm/s, the average strain rate is estimated to be 7.7×10-5 (1/s). In fact, the tension system was not “hard” and slip was present between the specimen and the grip, and thus V1 is smaller than V0. The above estimation is overestimated the real strain of the bulk material.

Let me first leave apart the question of the machine stiffness and consider a hard machine. The factor of 1/2 in V1/2 is wrong. Apparently, the authors think (erroneously) that the applied strain rate (imposed by the crosshead velocity) varies along the specimen, so that it should be averaged (more precisely, obtained by integration). It is not so. The mistake can be easily found if the problem is considered rigorously or by consulting a textbook on mechanics. Without going into detail, let me point out that although the end of the specimen is moved at a speed twice as high as that of the middle section, the length of the entire specimen which is elongated with this speed is also twice the length of the half specimen. The imposed strain rate is the same.  

Furthermore, the term "nominal applied strain rate" in Question 2 is a conventional term used to indicate the applied strain rate referred to the initial gage length. It is thus 0.01 mm/s \div 65 mm and is approximately equal to 1.5*10-4 s^-1. The reader will understand that this value gives the initial strain rate, while the real value is decreased during the test because of the specimen elongation. Anyway, the latter factor is negligible in this paper because only the beginning of the deformation is investigated, so that the cumulated strain is negligible.

This estimate, indeed, suggests a sufficiently hard machine. The authors claim that the machine is not hard enough and the estimate should be corrected downward. In this case, the stiffness of the machine and the required correction should be specified explicitly. Moreover, if according to the authors, the real strain rate must be reduced by an order of magnitude to take into account the insufficient stiffness, the results obtained should be disregarded and the entire investigation ought to be done again with a correct setup. Nevertheless, regarding the deformation curves presented in the manuscript and the abruptness of stress serrations, I suppose that the machine is not as soft and the estimate of 1.5*10-4 s^-1 is not far from the true value. This cannot be said about the value of 10-5 s-1 suggested by the authors.

  1. Methodological mistake.

Old question 3. “- Figure 3. The Lueders plateau is accompanied with stress serrations with an amplitude comparable with the difference between the upper yield stress and the estimate suggested by the authors for the lower yield stress. It cannot be said if the latter does exist. Very likely, it does not correspond to the lower yield stress in terms of the scheme of Figure 1 but to the unloading produced by the first stress drop. In other words, the upper and lower yield stress values may be equal for this material. This ambiguity in the possible interpretation of the data must be mentioned. Accordingly, it’d be correct to omit the tentative comparison in the paragraph starting from Line 171. This paragraph is even more so unclear because of the missing explanation of the method used to assess the local strain rate (see the above comment).

In this context, discuss the suggestion made in D. Yuzbekova et al, Intern. J. Plasticity 96 (2017) 210: ““The yield plateau is related to nucleation and fast propagation of a deformation band from one specimen edge to the other (Fig. 4), in agreement with the generally accepted Lueders mechanism (Schwab and Ruff, 2013; Antolovich and Armstrong, 2014). The first two frames seize the process of progressive nucleation of the band. This observation might explain the absence of a yield tooth that is known to often precede the Lueders plateau.” It is probable that the equality between the upper and lower yield stress for the studied material is related to such a progressive development of the Lueders band. Note that such a suggestion has two consequences:

- the fact of the early occurrence of the Lueders bands regarding the upper yield stress shouldn’t be considered as a rule but rather as one of the possible scenarios. The discussion and conclusions have to be rewritten in this sense.

- It should be underlined that future investigations are imposed using a material demonstrating a yield tooth.”

Authors’ answer: Two additional tests were performed and their results are shown in Figs. 7 and 8. The

scatter among the stress-strain curves in Fig. 4, Fig. 7(a) and Fig. 8(a) is great. In Fig. 1, the upper yield stress is larger than the stress at any point on the yield plateau. In this study, we denote the stress of the first peak on the yield plateau as the upper yield stress. The stress-strain curve in Fig. 8(a) is almost the same as that in Fig. 1. The upper yield stress in Fig. 4 is slightly larger than the stress on the yield plateau. However, the upper yield stress is not the largest on the yield plateau in Fig. 7(a). Because of stress fluctuation on the yield plateau, it is difficult to determine the lower yield stress. In this study, we took the average stress of the stress plateau as the lower yield stress. We performed tension tests on a model steel whose chemical composition is very clean (almost the same as the pure iron) and its microstructure is composed of uniform ferrite grains. Its stress-strain curve is pretty, almost completely the same as that in Fig. 1. There is almost no stress fluctuation on the stress plateau. However, in the present study, the stress fluctuation on the stress plateau was great.

The steel used in the study is a commercial steel. Its chemical composition is not clean. Its microstructure is composed of multiple phases (ferrite and pearlite), precipitates and inclusions. Because the steel was produced by hot-rolling, its microstructure along the plate thickness is not uniform. Even at the same thickness, ferrite grain size is also not uniform, and the pearlite prefers to distribute along rolling direction. The great stress fluctuation on the yield plateau in study is attributed to the great heterogeneity of microstructure.

The authors take the existence of a yield tooth as a dogma and give in to temptation to interpret the experimental data in this sense. I agree that it cannot be forbidden to use the value obtained by averaging over the plateau as an estimate of the lower yield stress for qualitative interpretations. But using it for the quantitative comparison at the end of page 7 is senseless. Note that this average is inevitably biased because of the presence of deep stress serrations. One could try to define the lower yield stress using some median value, instead on a mean, but let me attract the authors’ attention to the simple fact that both in Figs. 4 and 7 the upper yield stress is lower than numerous local stress maxima (i.e., multiple yield points). By the way, the affirmation that “The upper yield stress in Fig. 4 is slightly larger than the stress on the yield plateau” is incorrect not only for this reason. Starting from the strain about 0.008, the plateau shifts upward (the step is obvious), so that not only local maxima but also the median value exceed the “upper yield stress”. Only in Fig. 8, there is some significant difference between the first maximum and the subsequent maxima, so that the curve could be described by the scheme of Figure 1.

Therefore, the whole passage “Its experimental value is 394 MPa… verifying the proposed band nucleation criterion” should be omitted. Instead, it is necessary to discuss the possible mechanisms leading to the suppression of the yield tooth in the studied material. A possible answer may find itself in the complex microstructure of the commercial steel and the resulting heterogeneity of the yield in the region of the Lueders band nucleation. The authors speak of the microstructure complexity both in the above answer and in the new portion of the text on page 5 of the manuscript, but it is only evoked in order to interpret the occurrence of serrations on the plateau. I am surprised that although they mention tests on a more uniform material in the answer to reviewer, those are not mentioned in the manuscript. According to the sentence “Its stress-strain curve is pretty, almost completely the same as that in Fig. 1. There is almost no stress fluctuation on the stress plateau “, I guess that this model material showed a clear yield tooth. Note the end of the question in my first review: “It should be underlined that future investigations are imposed using a material demonstrating a yield tooth.” It occurs that such experiments have been already performed and a corresponding discussion can be added.  

The discussion of the suggestion in Yuzbekova’s paper is thus mandatory. Moreover, it may be developed in the following direction. Whereas the Yuzbekova’s paper suggests that the yield tooth does not show up if the Lueders band is developed progressively instead of being formed abruptly, it does not suggest a physical mechanism for the change between the two dynamic modes. The comparison between the commercial and model steel testifies that the former mode may be due to the strong heterogeneity of the microstructure in the commercial steel.

  1. Other corrections.

- Page 3, new portion of the text.  

Estimates of the constants k and n are should be given explicitly for the curve used for quantitative estimates.

- Same paragraph, sentence “The local stress was directly obtained from the DIC analysis, and its corresponding local stress was determined according to the basic stress-strain curve. “

The first occurrence of the word “stress” obviously must be replaced with “strain”.  

Figure 6 is unnecessary. It can be simply said that application of a rougher subset led to smoothening details of the pattern but provided qualitatively similar patterns, thus confirming the results of the analysis.  

End of page 7. The estimate of the local stress concentration at the onset of the Lueders band formation is 1.08. The number should be verified. If I understand well the definitions made by the authors, it must be equal to the ratio of the true upper stress to the observed upper stress, which was given as 1.07 in the portion to be omitted.

Author Response

Thank you for your comments.

Reviewer 2 Report

The authors have addressed all my remarks and criticisms. However, I found that their replies where quick and shallow. In particular, I think they did not address my main concerns. I also provide bellow few remarks to illustrate my point of view:

3) "Van Rooyen recognized the upper yield stress as a critical macroscopic parameter and suggested that when macroscopic applied stress reaches the upper yield stress, a Lüders band begins to form [4,5]. The upper yield stress is greater than the experimental upper yield stress [4,5]."

Why? And what the difference between these two stresses? Is it just a matter of stress concentration and inhomogeneities?

Answer: In Van Rooyen’s model, he only proposed an assumption regarding these two stresses. There is no experimental support. Therefore, in this study we try to verify this assumption by experiment.

I think this is not a convincing explanation.

4) It is written that :

"Schwab and Ruff [7] believed that the stress plateau in the nominal stress-strain curve does not truly reflect the material behavior."

Why it is the case? Once again the discussion lacks of depth.

Answer: In this stress plateau, there is elastic deformation region, and there is also plastic region (Lüder band), and thus the stress-strain curve is the mixture of both regions. Naturally, it does not truly reflect the material behavior.

This is non sens. Does it come from inhomogeneity in loading, in properties?

7) It is written that:

"The 2D-DIC measurement was within the global elastic region in which applied stress ranged from zero to σ up.ys.ob ."

Here again, I think there is problem in the sentence. Do the authors mean that the 2D-DIC measurement was calibrated, or tested, in the elastic regime?

Answer: Yes, the DIC measurement was performed in the macroscopic elastic regime.

I understand but the sentence does not mean anything. In addition, several DIC points reported in the manuscript (e.g. 3 and 4 from figure 8) are not in the global elastic regime.

11) "Figure 6(a) shows that the global applied stress level of 0.70σ up.ys.ob is an elastic- to-plastic threshold value, below which both the bulk material and the local region completely elastically deform and above which some plastic local regions gradually appear.”

On the basis of the figure, it is far to be obvious to me. One would have say 0.9 rather the

0.7. Can the authors justify their choices?

Answer: It is an experimental result. Below 0.70σup.ys.ob, even stress concentration is present, this local site is still in the elastic regime. At 0.9σup.ys.ob, although the bulk material is in the macroscopic elastic region, microyielding has taken place in some local sites.

I think that the authors still have to justify it.

13) "The front of the Lüders band inclined 55°in this study (see Fig. 4(a)- 4 ). For this angle, the Schwab-Ruff model gives σ up.ys.tr /σ L.ys.tr =1.26 and σ L.ys.ob /σ L.ys.tr =1.16, deducing that σ up.ys.tr =1.09σ L.ys.ob . According to the Schwab-Ruff model, σ up.ys.tr /σ up.ys.ob = 1.09σ L.ys.ob /σ up.ys.ob . Substituting the experimental values,giving σ up.ys.tr /σ up.ys.ob =1.05. The model-estimated σ up.ys.tr /σ up.ys.ob (=1.05) agrees with the experimental one (=1.07), verifying the proposed band nucleation criterion."

The model is absolutely not introduced or discussed.

Answer: The model is well known in this aspect. Moreover, the aim of this study is not to verify this model.

I do not mean to verify the model but to properly introduce it and discuss its outcomes and limits, and put it in a comprehensible perspective with respect to the present work. I think this is not the case here.

14) The authors write:

"The plastic strain at the onset of band nucleation in this study was 365 , much greater

than Vreeland’s data. Vreeland probably underestimated the real plastic strain because of

the imprecision measurement technique at that time."

In the absence of further analysis, this proposition is truly speculative. Answer: Strain gage was used to measure strain in Vreeland’s work. It is known that strain gage can only measure the average strain over the region (strain gage size), cannot measure local strain. Naturally, his data underestimated the true local strain.

I understand. As a consequence, given the accuracy of the DIC technique use in the present work, one should be able to discuss on a more quantitative basis the order of magnitude of this underestimation.

Author Response

Thank you for your comments.

Reviewer 3 Report

The authors have done a good job addressing the reviewers' comments. 

Author Response

Thank you for your comments.

Round 3

Reviewer 1 Report

The authors have answered to my questions. Even if some aspects of the results interpretation could have been further clarified, the paper is worth being presented to a large audience. I recommend it for publication.